A randomized clinical trial of vitamin D3 (cholecalciferol) in ulcerative colitis patients with hypovitaminosis D3

Mathur Jagrati 1
Naing Soe 2
Mills Paul 3
Limsui David dlimsui@stanford.edu 4
1 Division of Gastroenterology and Hepatology, University of California, San Francisco , Fresno , CA , United States of America
2 Division of Endocrinology, University of California, San Francisco , Fresno , CA , United States of America
3 Department of Epidemiology, University of California, San Francisco , Fresno , CA , United States of America
4 Division of Gastroenterology and Hepatology, Stanford University School of Medicine , Stanford , CA , United States of America
Seccia Teresa
Electronic publication date: 2017 Aug 3
Publication date: 2017
Volume: 5
Electronic Location ID: e3654
Received 2017 Feb 6; Accepted 2017 Jul 14
Copyright: ©2017 Mathur et al.
Copyright year: 2017
Copyright holder: Mathur et al.
License: This is an open access article distributed under the terms of the Creative Commons Attribution License, which permits unrestricted use, distribution, reproduction and adaptation in any medium and for any purpose provided that it is properly attributed. For attribution, the original author(s), title, publication source (PeerJ) and either DOI or URL of the article must be cited.
License URL: https://creativecommons.org/licenses/by/4.0/

Keywords: Ulcerative colitis, Vitamin D, Quality of life, Disease activity

Funding: Central California Faculty Medical Group in Fresno, California This work was supported by a grant received from the Central California Faculty Medical Group in Fresno, California. The funders had no role in study design, data collection and analysis, decision to publish, or preparation of the manuscript.

==============================
Aim

To prospectively evaluate the effects of vitamin D3 on disease activity and quality of life in ulcerative colitis (UC) patients with hypovitaminosis D.

Methods

The study was a prospective double-blinded, randomized trial conducted at Community Regional Medical Center, Fresno, CA from 2012–2013. Patients with UC and a serum 25(OH)D level <30 ng/ml were eligible for the study. Enrolled subjects were randomized to receive either 2,000 IU or 4,000 IU of oral vitamin D3 daily for a total of 90 days. The Short IBD Questionnaire (SIBDQ) for quality of life, the Partial Mayo Score for UC disease activity and serum lab tests were compared between the two treatment groups. Matched pair t-tests were computed to assess differences between the vitamin D levels, CRP, UC disease activity and SIBDQ scores before and after vitamin D3 therapy using SPSS version 21.

Results

Eight UC patients received 2,000 IU/daily and ten UC patients received 4,000 IU/daily of vitamin D3 for 90 days. Vitamin D levels increased after 90 days of oral vitamin D3 in both dose groups. However, the increase in vitamin D levels after 90 days of oral vitamin D3, in the 4,000 IU group was significantly higher 16.80 ± 9.15 (p < 0.001) compared to the 2,000 IU group of vitamin D 5.00 ± 3.12 (p = 0.008). Normal vitamin D levels (>30 ng/dl) were achieved in four out of the ten UC patients (40%) in the 4,000 IU group and in one out of the eight UC patients (12%) in the 2,000 IU group. In the group receiving 4,000 IU/day of vitamin D3 the increase in quality life scores (SIBDQ) was significant 1.0 ± 1.0 (p = 0.017) but not in the 2,000 IU vitamin D3 group 0.1 ± 1.0 (p = 0.87). In the 2,000 IU of vitamin D3 group the mean decrease in the Partial Mayo UC Score was −0.5 ± 1.5 (p = 0.38) compared to −1.3 ± 2.9 (p = 0.19) in the 4,000 IU vitamin D3 group but this was not statistically significant. CRP levels decreased after 90 days of daily vitamin D3 in both the 2,000 IU group and 4,000 IU group by −3.0 ± 9.4 (p = 0.4) and −10.8 ± 35.0 (p = 0.36) respectively.

Conclusion

Vitamin D3 at 4,000 IU/day is more effective than 2,000 IU/day in increasing vitamin D to sufficient levels in UC patients with hypovitaminosis D, however higher doses or treatment beyond ninety days may be required. Vitamin D3 may improve the quality of life in UC patients but clinically significant improvement is not yet established. The effect of vitamin D3 on UC disease activity is still unclear. Further larger studies are needed to investigate the effects of vitamin D in ulcerative colitis.

Introduction

Ulcerative colitis (UC) is an idiopathic, chronic, immune mediated disease that affects the gastrointestinal tract. It results from an exaggerated host immune response to luminal antigens or intestinal microflora in the gastrointestinal tract causing inflammation and has a relapsing and remitting course (Lim, Hanauer & Li, 2005; Ananthakrishnan, 2015). What triggers this exaggerated immune response is not clearly understood and is thought to be due to a complex interplay of genetic, environmental, immune and microbial factors (Podolsky, 2002). The concordance estimates of UC between twins is 20% or less, suggesting non-genetic factors also play a significant role in the pathogenesis of UC (Cho & Abraham, 2007). Some environmental factors that have been associated with IBD include cigarette smoking, oral contraceptive use, vitamin D levels, dietary factors, stress, non-steroidal anti-inflammatory drugs, physical activity and duration of sleep (Ananthakrishnan, 2015). Vitamin D is increasingly being identified as an important environmental factor influencing many chronic diseases including cancers, cardiovascular disease, and autoimmune diseases such as multiple sclerosis and IBD (Ulitsky et al., 2011).

Vitamin D is a fat soluble vitamin and its role in maintaining and promoting bone health by increasing intestinal absorption of calcium is well known. However, the role of vitamin D as an immunomodulator is now being increasingly recognized. The first evidence of vitamin D as an immunomodulator came in 1983 with the discovery of vitamin D receptors (VDR) on various immune cells (Cantorna et al., 2004). Vitamin D can down regulate the inflammatory cascade by decreasing the proliferation of T cells (Th1), inhibiting cytokine production of IL-2 and INF-V and inducing proliferation of regulatory T cells (Cantorna & Mahon, 2004). 1,25 (OH)2D3 also inhibits differentiation of peripheral blood monocytes into dendritic cells (Lim, Hanauer & Li, 2005). The prevalence of vitamin D deficiency in UC patients ranges from 45–60%. (Pappa et al., 2006). Animal experiments, epidemiologic and genetic studies have suggested that vitamin D levels may influence IBD. Further large population studies including the prospective Nurses Health Study have shown that increased vitamin D intake was associated with a reduced incidence of IBD (Ananthakrishnan et al., 2012). In a cross-sectional study of UC patients, vitamin D deficiency was independently associated with higher disease activity scores and increased steroid use than patients with sufficient vitamin D levels (Blanck & Aberra, 2013). Low vitamin D levels have also been associated with lower quality of life (QOL) in patients with IBD (Ulitsky et al., 2011).

Despite the evidence suggesting an association between vitamin D and IBD, there have been no prospective human studies looking at the effects of vitamin D3 in patients with ulcerative colitis. The appropriate dose of vitamin D3 in the management of IBD and hypovitaminosis D (vitamin D levels <30 ng/ml) is also not well understood.

The primary purpose of this pilot study was to examine prospectively the effects of two different doses of vitamin D3 on vitamin D levels in UC patients with hypovitaminosis D to help determine the appropriate dose for this patient population. The secondary aim was to examine the effects of two different doses of vitamin D3 on disease activity and quality of life in UC patients with hypovitaminosis D. Our hypothesis was that oral vitamin D3 doses of 2,000 IU and 4,000 IU daily would increase vitamin D levels in both groups but the change would be greater in the 4,000 IU group. We also hypothesized that oral vitamin D3 could be associated with a decrease in disease activity scores and increase in quality of life scores in patients with UC and hypovitaminosis D.

Methods

The Community Medical Centers Institutional Review Board granted approval to carry out this study within its facilities. (IRB number 2012043) The study was a prospective double-blinded, randomized pilot trial conducted at Community Regional Medical Center (CRMC) in Fresno, California from June 2012–July 2013. Patients with ulcerative colitis and serum 25(OH) vitamin D levels less than 30 ng/ml within a year of enrollment, were eligible for participation in the study. Exclusion criteria included age less than eighteen years, pregnant females and patients on vitamin D supplementation >2,000 IU/day. All study participants signed a written voluntary informed consent document approved by the IRB prior to their enrollment in the study.

Ulcerative colitis patients at CRMC are routinely screened for vitamin D deficiency by checking serum 25(OH) vitamin D levels. Interested and eligible patients were referred by their providers to the research coordinator and co-investigators of the study for possible enrollment and were confirmed to meet eligibility criteria. Participants were required to come for two study visits—first at the time of enrollment and next at the end of the ninety-day study period. During the initial visit each participant completed a data collection form including information regarding: age, sex, medical history, duration and location of UC, smoking history, current medications, sun exposure, and last corticosteroid use. At both study visits, participants completed two questionnaires together with the investigators and had serum blood drawn for laboratory studies. The two questionnaires included the Partial Mayo Score (PMS) to determine UC disease activity and the short IBD quality of life questionnaire score (SIBDQ) to measure quality of life in UC patients. The validated Partial Mayo Score for UC disease activity includes sub-scores ranging from 0–3 in the following categories: stool frequency, rectal bleeding and physician’s global assessment of well-being (Lewis et al., 2008). The Partial Mayo Score is the sum of these three sub-scores and ranges from zero which is normal and up to nine for severe disease. The validated ten item SIBDQ includes questions based on bowel, systemic, emotional and social domains with each question score ranging from 1–7. The total SIBDQ score is calculated by adding the total points of the ten items and then dividing it by ten with one being the lowest quality of life and seven being the highest quality of life (Irvine, Zhou & Thompson, 1996). Serum blood draws for laboratory studies were completed at both study visits and included: a complete blood count (CBC) with differential, liver function tests, renal function test, serum phosphorous, serum calcium, serum parathyroid hormone (PTH), erythrocyte sedimentation rate (ESR) and C-reactive protein (CRP) and 25(OH) vitamin D.

Each study participant was randomized to receive 2,000 IU/daily or 4,000 IU/daily of vitamin D3 for ninety days (Fig. 1). The ninety day supply of vitamin D3 for each subject was relabeled and packaged so that both the subjects and the investigators were blinded throughout the study. The packages were distributed by block-randomization among the study participants as they enrolled into the study and a record was kept by a randomization list. The sealed envelope with the randomization list was never opened by the investigators to unmask blinding until study completion. Medication compliance was evaluated through patient interview and by counting the number of pills left at the final visit. Adverse events of vitamin D were monitored for and participants were asked about any possible side effects associated with vitamin D toxicity during the study visits and with a follow up phone call at day forty-five.

Figure 1 Study overview.

Statistical Methods

Funding limited the sample size and enrollment of this pilot study to twenty subjects. We anticipated that there would be an improvement in vitamin D levels in both groups. However, we hypothesized that the increase in vitamin D would be greater in the 4,000 IU group vs the 2,000 IU group. We estimated that the change in the vitamin D levels in the 4,000 group would be about 30% greater than in the 2,000 IU group. Thus using the traditional two-sided confidence interval of 95% with 10 patients in the 4,000 IU group and 10 patients in the 2,000 IU group, we calculated that the study would achieve a power of 28%. We recognized the limitations of this low power but this was a pilot study. Partial Mayo Score for UC disease activity, SIBDQ score and serum lab values for 25(OH) vitamin D and CRP before and after ninety days of vitamin D3 treatment were compared and analyzed between the 2,000 IU and 4,000 IU groups at the end of the study. The correlation between change in 25(OH) vitamin D levels and change in disease activity and quality of life scores were also compared across all study subjects using Pearson correlation coefficient. All tests were two-sided with a p value of p < 0.05 considered as statistically significant. Continuous variables were compared using Student’s t-test and categorical variables were compared using Fisher’s exact test. Matched pair t-test were computed to assess differences between the vitamin D levels, CRP, UC disease activity scores and SIBDQ scores before and after vitamin D3 supplementation using SPSS version 21.

Results

Twenty patients with ulcerative colitis and hypovitaminosis D were enrolled during the study period. Two patients were excluded from the study: one patient had normal vitamin D levels at the time of randomization, and one patient left the study without taking their vitamin D3 treatment. Thus, a total of 18 patients completed the study and were included in the analysis of results which was therefore not an intention to treat analysis. Eight patients received 2,000 IU of vitamin D3 and ten patients received 4,000 IU of vitamin D3 for ninety days and were followed prospectively.

At study enrollment and at baseline there were no significant differences in sex, age, smoking status, BMI, UC duration history, daily sun exposure and use of anti-TNF and immunomodulator therapies between the two treatment groups. Baseline serum levels of 25(OH) vitamin D, ESR, CRP, PTH, calcium, albumin, and phosphorus and SIBDQ quality of life scores were also similar between the two groups. The baseline Partial Mayo Score for UC disease activity was significantly higher in the 4,000 IU vitamin D3 group compared to the 2,000 IU vitamin D3 group, with a mean score of 4.0 versus 1.4 respectively (Table 1). None of the patients enrolled in the study had a colectomy.

Table 1 Baseline characteristics.

Characteristics	Vitamin D3 2,000 IU/day (n = 8)	Vitamin D3 4,000 IU/day (n = 10)	p value	
Sex, n (%)			0.2	
Male	7 (88)	6(60)		
Female	1(13)	4(40)		
Age, years, mean (SD)	41.1 (13.7)	40.2 (16.2)	0.9	
Ethnicity, n (%)				
White	4 (50)	5 (50)		
Hispanic	3 (38)	3 (30)		
African American	1 (13)	0 (0)		
Asian	0 (0)	2 (20)		
Current smoker, n (%)			0.44	
Yes	1(13)	0 (0)		
No	7 (88)	10 (100)		
BMI, mean (SD)	27.78 (4)	25.7(5.4)	0.32	
UC duration, y, mean (SD)	3.7 (2.9)	5.2 (4.7)	0.57	
Daily hours of sun, mean (SD)	3.6 (2.4)	2.4 (2.5)	0.22	
Baseline vitamin D level (ng/ml), mean (SD)	17 (5.24)	14.3 (4.24)	0.24	
Partial Mayo UC disease activity score, mean (SD)	1.4 (1.2)	4.0 (2.3)	0.03	
SIBDQ score, mean (SD)	5.6 (0.6)	4.8 (1.4)	0.21	
ESR (mm/hr) (normal: 0–10), mean (SD)	10.5 (8.4)	27.0 (26.7)	0.2	
CRP (mg/dl) (normal: 0–3), mean (SD)	6.9(8.8)	20.8 (34.8)	0.21	
PTH (pg/ml) (normal: 14–72), mean (SD)	42.7(12.7)	46.4 (15.8)	1	
Calcium (mEq/L) (normal: 8.5–10.5), mean (SD)	9.5(0.5)	9.3 (0.5)	0.6	
Albumin (g/dl) (normal: 3.4–4.8), mean (SD)	4.5 (0.3)	4.2 (0.6)	0.66	
Phosphorous (mg/dl) (normal: 2.7–4.5), mean (SD)	3.2 (0.7)	3.7 (0.9)	0.38	

During the course of the entire study patients remained on their current medications for treatment of ulcerative colitis. At the time of enrollment, three of the UC patients were on anti-TNF medications, one patient was on azathioprine and the remaining fourteen patients were on 5-aminosalicylic acid medications. There were no changes to these medications and no other new medications started during the study period. No study patients were hospitalized or required surgery during the study.

Vitamin D levels increased after ninety days of oral vitamin D3 in both treatment groups. However, the increase in vitamin D level in the 4,000 IU group of 16.8 ± 9.15 (p < 0.001) was significantly higher compared to the increase in the 2,000 IU group of 5.00  ±  3.82(p = 0.008). (Fig. 2). Normal vitamin D levels (>30 ng/ml) were achieved in four of ten (40%) of UC patients in the 4,000 IU group and in one of eight (12%) of UC patients in the 2,000 IU group after ninety days of vitamin D3 treatment. Quality of life scores as measured by SIBDQ increased among all the UC patients after ninety days of vitamin D3. This increase in SIBDQ scores was significant in the 4,000 IU vitamin D3 group 1.00 ± 1.00 (p = 0.017) but not in the 2,000 IU vitamin D3 group 0.10 ± 1.00.

Figure 2 Change in vitamin D levels after ninety days.

Overall UC disease activity scores decreased in both treatment dose groups after ninety days of vitamin D3; however, this change was not significant. In the group receiving 2,000 IU of vitamin D3 daily, the mean decrease in the Partial Mayo Score was −0.5 ± 1.5 compared to −1.3 ± 2.9 in the 4,000 IU group. Similarly, CRP levels overall decreased in both treatment dose groups after ninety days of vitamin D3; however, this change was not significant. CRP level decreased by 3.0 ± 9.4 in the group receiving 2,000 IU of vitamin D3 and decreased by 10.0 ± 35.0 in the 4,000 IU group.

Across all eighteen UC patients in the study, the increase in vitamin D levels after ninety days correlated with an increase in SIBDQ quality of life scores with a Pearson correlation coefficient of 0.52, p = 0.028, and this increase in vitamin D level also correlated with a decrease in Partial Mayo UC activity scores at −0.58, p = 0.012. A two-sample independent t-test was used to compare the mean changes in the primary and secondary end points between the two treatment groups after ninety days.

(a) The mean change in the vitamin D levels between the two treatment groups was significant at p < 0.003.

(b) The mean change in the quality of life between the two treatment groups was not significant.

(c) The mean change in the disease activity scores between the two treatment groups was not significant.

(d) The mean change in the CRP levels between the two treatment groups was not significant.

Medication compliance was evaluated through patient interview and by counting the number of pills left at the final visit. All eighteen patients were adherent with their oral vitamin D3. Adverse events of vitamin D were monitored by asking the participants about any possible side effects associated with oral vitamin D toxicity during the study visits and at day forty-five. Oral vitamin D3 was well-tolerated by all the study participants with no adverse events reported throughout the study. Serum calcium and parathyroid hormone levels were checked during the study and were normal. There were no signs or symptoms of vitamin D toxicity observed in either dose groups.

Discussion

This is the first prospective randomized pilot study to our knowledge evaluating the effects of vitamin D3 on the disease activity and quality of life in patients with UC. The prevalence of vitamin D deficiency in adults with ulcerative colitis has been reported to be as high as about 45–50% and has been attributed to various factors including decreased sunlight exposure, low oral vitamin D intake, disturbed enterohepatic circulation and increased loss of vitamin D as the result of protein-losing enteropathy. (Ulitsky et al., 2011). However, it is unclear if vitamin D deficiency is an environmental trigger for autoimmunity in IBD or if IBD directly causes vitamin D deficiency.

This study evaluated the effects of oral vitamin D3 on QOL after ninety days in patients with UC and hypovitaminosis D. The results show that UC subjects receiving 4,000 IU/day of vitamin D3 had significantly higher quality of life scores after ninety days as compared to the 2,000 IU/day group. However, the mean increase in SIBDQ of 1.00 in the 4,000 IU/day treatment group of our study is unlikely clinically significant when compared to a previously reported mean reduction of 11.8 in SIBDQ being associated with change in UC disease activity from remission to relapse. (Jowett et al., 2001). This may suggest that higher doses of oral vitamin D3 may be needed to achieve beneficial vitamin D levels and higher QOL in UC patients.

Vitamin D deficiency in IBD patients has also been shown to be an independent risk factor for higher disease activity scores and increased frequency of corticosteroid use (Blanck & Aberra, 2013; Ulitsky et al., 2011). Vitamin D affects both the innate and adaptive immune systems and leads to immune-tolerance of self-structures. (Ardizzone et al., 2011). Vitamin D has been used in other autoimmune diseases such as multiple sclerosis, rheumatoid arthritis, and systemic lupus erythematous and found to have a beneficial effect in reducing disease severity. Vitamin D through its receptors influences the maturation and differentiation of antigen presenting cells, dendritic cells and macrophages, resulting in the decreased activation of T cells and suppression of inflammatory cytokines (Lim, Hanauer & Li, 2005) and may thereby reduce disease activity in IBD.

In this study oral vitamin D3 for ninety days in UC patients decreased UC disease activity scores with a mean decrease in Partial Mayo Score of 0.5 in the 2,000 IU/day group and 1.3 in the 4,000 IU/day group, but this failed to reach statistical significance. The decrease in disease activity score was also unlikely clinically significant when compared to a previously reported mean decrease of 2.5 in the Partial Mayo Score being associated with patient perceived clinical response (Lewis et al., 2008). A possible explanation for this could be because only 40% of UC patients (4 of 10) in the 4,000 IU/day group and 12% (1 of 8) in the 2,000 IU/day group achieved sufficient vitamin D levels of >30 ng/ml after ninety days. This may suggest that a longer duration or higher doses of vitamin D are required to achieve sufficient vitamin D levels >30 ng/ml to affect UC disease activity. However, it is also possible that continued active UC is causing hypovitaminosis D and preventing achievement of sufficient vitamin D levels. It has also been suggested that there may be two levels of vitamin D that could be relevant to the management of IBD: vitamin D levels less than <20 ng/ml that are associated with higher risk for developing osteoporosis, osteomalacia or rickets and vitamin D levels >30 ng/ml which may ensure normal immune system regulation (Lim, Hanauer & Li, 2005).

In this study, increases in vitamin D levels correlated with higher QOL scores and decreased UC disease activity scores in UC patients after ninety days of oral vitamin D3. Other studies have shown normal vitamin D levels can predict a decreased incidence risk of IBD. (Ananthakrishnan et al., 2012). Similarly lower vitamin D levels have been reported in newly diagnosed children with IBD (El-Matary, Moroz & Bernstein, 2014).

To date there is no consensus on the appropriate dosage of vitamin D needed in patients with UC with vitamin D deficiency. This pilot study compared the effects of two different doses of vitamin D3 with 2,000 IU/daily versus 4,000 IU/day in UC patients. Patients with UC who received 4,000 IU/day of vitamin D3 for ninety days had significantly higher vitamin D levels and increased quality of life scores compared to the 2,000 IU/day group. Because of the small sample size no further conclusion can be drawn. Another limitation of the study was the lack of a placebo control group, which was not feasible with the study’s IRB noting that appropriate care would be to treat all UC patients with hypovitaminosis D with vitamin D replacement therapy. This study also does not include endoscopic evaluation for mucosal healing and potential dietary sources of vitamin D may be a confounding variable.

We conclude from this study that vitamin D3 at a higher dose of 4,000 IU/day is more effective than a lower dose of 2,000 IU/day in increasing vitamin D to sufficient levels in UC patients with hypovitaminosis D. Further larger and longer-term studies are needed to investigate the effects of vitamin D in ulcerative colitis.

Supplemental Information

Supplemental Information 1 Vitamin D in UC

Click here for additional data file.

Supplemental Information 2 Study protocol

Click here for additional data file.

Supplemental Information 3 CONSORT checklist

Click here for additional data file.

Additional Information and Declarations

Competing Interests

Author Contributions

Clinical Trial Ethics

Data Availability

Clinical Trial Registration

The authors declare there are no competing interests.

Jagrati Mathur conceived and designed the experiments, performed the experiments, analyzed the data, wrote the paper, prepared figures and/or tables, reviewed drafts of the paper.

Soe Naing conceived and designed the experiments, analyzed the data, reviewed drafts of the paper.

Paul Mills conceived and designed the experiments, analyzed the data, contributed reagents/materials/analysis tools, reviewed drafts of the paper.

David Limsui conceived and designed the experiments, performed the experiments, analyzed the data, wrote the paper, reviewed drafts of the paper.

The following information was supplied relating to ethical approvals (i.e., approving body and any reference numbers):

The Community Medical Centers Institutional Review Board granted approval to carry out the study within its facilities (IRB Number 2012043).

The following information was supplied regarding data availability:

The raw data has been supplied as a Supplementary File.

The following information was supplied regarding Clinical Trial registration:

NCT 01877577.

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
