# Peer review of "A randomized clinical trial of vitamin D3 (cholecalciferol) in ulcerative colitis patients with hypovitaminosis D3"

_PeerJ, doi:10.7717/peerj.3654_

## Round 0.1 · original submission · Major Revisions

· Academic Editor

Major Revisions

As suggested by the Reviewers, some methodological and statistical issues need attention. I hope that the Reviewers' comments help you with the revision.

Reviewer 1 ·

Basic reporting

1. There are minor typographical errors:
a. Line 83 – “chronic disease” should be plural
b. Lines 87-90 – sentence is too wordy and could potentially be made clearer by eliminating some of the words.
2. Overall the paper is well written, but it could be better organized, with certain topics expanded upon and others made more concise.
a. Lines 244-247 regarding safety and tolerability should be moved to results section.
b. Introduction is probably somewhat lengthier than needed, yet despite the use of HRQOL as an outcome measure, the importance of vitamin D to HRQOL in UC is not mentioned until the discussion section (reference 25, lines 219-221). If anything this sentence, and possibly the one preceding, should be moved from the discussion section to the introduction and the rest of the introduction made more succinct.
c. Figure 5 does not add much information beyond what is already in the text. It seems redundant.
3. With regards to "relevant results to hypotheses," I do not believe the authors state their hypotheses in the article.
4. The inclusion of Figures 3 and 4 (graphical displays of changes in SIBDQ and partial Mayo score) does not really add much to the text especially in light of the fact that these changes are probably not clinically meaningful (see section of review on Validity of the findings). I would consider removing these figures.
5. The authors should discuss the clinically meaningful differences that are associated (for example) with different levels of disease activity to give their results a context. For example, the authors could cite the paper by Jowett et al. from Am J Gastro 2001;96:2921-8 which found that a mean reduction in the SIBDQ of 11.8 points was associated with a change in disease activity. Likewise, the Lewis paper cited by the authors on the partial Mayo score reports that a cutpoint of 2.5 for the partial Mayo score can be used to identify patient perceived clinical response.

Experimental design

1. The authors should specify the primary and secondary aims of the study. Was the primary aim to determine the appropriate dose of vitamin D3 for UC patients with hypovitaminosis D? It is understood that the outcomes included vitamin D levels, disease activity, and quality of life, but specifying the primary aim will add clarity to the study. Likewise, the authors should describe the specific associated hypothesis being tested.
2. Statistical methods section needs to be more detailed.
a. Although the sample size was limited by funding, it would be helpful to report the power that the planned study size had to detect for the primary endpoint.
b. Paired t-tests are not mentioned in the statistical methods section – were these used to compare changes in vitamin D levels after supplementation?

Validity of the findings

1. The measures used to assess quality of life and disease activity are associated in the literature with specific clinically meaningful differences. These differences are not discussed by the authors; rather, it appears that the authors are looking specifically at statistical significance, which may or may not be clinically meaningful. Therefore, the authors’ conclusion in the abstract and results that the SIBDQ increase in the 4,000 IU/day group was statistically significant needs to be qualified with an acknowledgment that this change was likely not clinically significant. As a corollary, the conclusions in the abstract and article that vitamin D3 may improve quality of life and disease activity in UC patients with hypovitaminosis D are too strong a statement.
2. Greater detail needs to be provided regarding the subjects’ disease course while participating in the study, since this would impact HRQOL and partial Mayo score. Were any subjects hospitalized or treated for UC flare during their participation in the study?
3. The authors do not mention pill counts other than to state that one patient did not take any vitamin D. They should explain whether the rest of the participants were adherent.
4. Since results from the subject not taking vitamin D were not included this is not an intention to treat analysis. That should be mentioned.
5. Table 1 and the text of the article present vitamin D levels in ng/mL, yet Figure 2 shows changes in vitamin D levels in ng/dL. Is this a typographical error?

Reviewer 2 ·

Basic reporting

I am a little confused as to why the P values are limited to within groups rather than between the 2 treatment groups. There are no P values presented comparing outcomes between the 2000 and 4000 doses on any of the Figures.

Experimental design

This is an important and timely topic though the study has substantial limitations:
1. There is no discussion of a sample size calculation. Unfortunately, the sample size was determined by the budget, but I would have liked to see what difference was expected or could have been demonstrated based on a power calculation of the primary outcome, which was not defined specifically.
2. Again, I imagine due to budget constraints, the partial Mayo score was used to assess response to vit D but this scoring system is subjective. Clearly, mucosal healing would be preferred to determine if there was really a response or not.
3. It is odd that there are no inclusion criteria based on active disease. Were patients recruited with no regard to whether they had active disease? What would it mean clinically to have a patient in remission but with hypovitaminosis D, and should such a patient be excluded?
4. There is no mention as to what medications the subjects were on at the time of enrollment and whether any change of medication was allowed during the treatment phase.
5. Clearly, it would have been much more informative and helpful with regard to the power of the study if there had been a placebo arm. I am not surprised that the IRB concluded that there should not be a placebo arm, but this type of conservative approach does not help us answer the important questions in science, including the question asked in this study. Since, to date, there is absolutely no evidence that administering Vit D to IBD patients independently provides a therapeutic benefit, it is not necessarily unethical to conduct a trial using a placebo in my opinion. Nevertheless, another study design could have both appeased the IRB and led to a more convincing conclusion, e.g. cross-over design, 2:1 ratio of vit D to placebo, or open label Vit D after placebo phase.

Validity of the findings

Based on this study, it is purely speculative to comment in the discussion that a longer period of Vit D supplementation would result in a "sufficient" level of Vit D, especially since it is possible that continued active disease may not allow this to occur.

Comments for the author

none

---

## Round 0.2 · Minor Revisions

· Academic Editor

Minor Revisions

I read carefully your manuscript and found the changes suggested by the Reviewers. However, my opinion is that this manuscript reports the findings of a very small study designed with several major limitations.

I recommend the Authors make the following changes before reconsidering the manuscript for publication:

1. To modify the description of the Results at pages 7-8 (following the tracked ms).
E.g.: 'The mean change in the vitamin D levels between the two treatment groups was statistically significant at p <0.003'. Statistically is redundant.
Moreover, please do not report p value if not significant. It is meaningfulness.

2. To modify the tables (please see the layout of tables reported in journals with high IF as example). Remove a), b), ... when listing variables, and also vertical lines. Please make the tables more readable.

3. Please change the Figure 1 (see the flow chart in the major journals as example). It is not acceptable in the present format.

---

## Round 0.3 · Minor Revisions

· Academic Editor

Minor Revisions

Please see the attached file with corrections.

---

## Round 0.4 · accepted · Accept

· Academic Editor

Accept

Dear Author,

I am writing to inform you that your manuscript - A randomized clinical trial of vitamin D3 (cholecalciferol) in ulcerative colitis patients with hypovitaminosis D3 - has been Accepted for publication.